# Bazedoxifene, a GP130 Inhibitor, Modulates EMT Signaling and Exhibits Antitumor Effects in HPV-Positive Cervical Cancer

**DOI:** 10.3390/ijms22168693

**Published:** 2021-08-13

**Authors:** Leekyung Kim, Sun-Ae Park, Hyemin Park, Heejung Kim, Tae-Hwe Heo

**Affiliations:** Laboratory of Pharmacoimmunology, Integrated Research Institute of Pharmaceutical Sciences and BK21 FOUR Team for Advanced Program for SmartPharma Leaders, College of Pharmacy, The Catholic University of Korea, 43 Jibong-ro, Bucheon-si 14662, Gyeonggi-do, Korea; 1202dlruddl@catholic.ac.kr (L.K.); tjsdo37@catholic.ac.kr (S.-A.P.); okpc5094@naver.com (H.P.)

**Keywords:** Bazedoxifene, HPV-positive, cervix cancer, EMT signaling

## Abstract

Persistent HPV (Human Papillomavirus) infection is the primary cause of cervical cancer. Despite the development of the HPV vaccine to prevent infections, cervical cancer is still a fatal malignant tumor and metastatic disease, and it is often difficult to treat, so a new treatment strategy is needed. The FDA-approved drug Bazedoxifene is a novel inhibitor of protein–protein interactions between IL-6 and GP130. Multiple ligand simultaneous docking and drug repositioning approaches have demonstrated that an IL-6/GP130 inhibitor can act as a selective estrogen modulator. However, the molecular basis for GP130 activation in cervical cancer remains unclear. In this study, we investigated the anticancer properties of Bazedoxifene in HPV-positive cervical cancer cells. In vitro and in vivo experiments showed that Bazedoxifene inhibited cell invasion, migration, colony formation, and tumor growth in cervical cancer cells. We also confirmed that Bazedoxifene inhibits the GP130/STAT3 pathway and suppresses the EMT (Epithelial-mesenchymal transition) sub-signal. Thus, these data not only suggest a molecular mechanism by which the GP130/STAT3 pathway may promote cancer, but also may provide a basis for cervical cancer replacement therapy.

## 1. Introduction

Cervical cancer is the fourth most common cancer and second most common gynecological cancer in the global female population [1,2,3]. According to the International Agency for Research on global cancer estimation, there were 569,847 new cervical cases and 311,365 deaths worldwide in 2018, accounting for 3.2% of new female cancer cases and deaths [4]. Although extensive cytological examination has dramatically reduced the morbidity and mortality of cervical cancer in many developed countries, it still remains a serious health risk [5].

IL-6 plays a role in recruiting immune cells within the tumor microenvironment and stimulates the production of pro-inflammatory cytokines. It is also involved in the pathogenesis of several chronic inflammatory diseases and promote the development and progression of tumors. IL-6 can be expressed at high levels in the tumor microenvironment and is a major mediator of inflammation. In addition, IL-6 can directly stimulate the proliferation, survival, and invasiveness of tumor cells. Thus, IL-6 serves as a key link between chronic inflammation and tumor progression [6]. IL-6 is produced by cancer cells, as well as inflammatory and stromal cells. Several studies have investigated the ability of IL-6/GP130 to promote chemotherapy resistance in several cancers as IL-6 is known to bind to the extracellular surface receptor glycoprotein 130 (GP130) and activate cell survival-related pathways [7,8,9]. GP130 is a receptor for IL-6 and a major intracellular signaling molecule. Currently, three signaling pathways are known to be associated with GP130, the JAK signaling and transcriptional activation (STAT) pathway, the ERK pathway, and the PI3K/AKT pathway. The most prominent proteins recruited to GP130 are the STAT family transcription factors STAT3 and (to a certain extent) STAT1. Moreover, it is now well known that STAT3 and (to a much lesser extent) STAT1 are activated by IL-6. Binding to IL-6 triggers phosphorylation of GP130, activating the cytoplasmic region [10,11]. Signal transducer and activator of transcription (STAT) 3 is an essential regulator of cellular proliferation, differentiation, and survival [12]. STAT3 is activated by oncogenic viruses and can promote the viral replication and cell proliferation required for tumor formation [13]. STAT3 is a bona fide oncogene and has been observed to become active in many malignancies and through phosphorylation of on STAT3 (P-STAT3) to exert its tumor-promoting effects [14,15]. Therefore, STAT3 has become a therapeutic target in a variety of cancers, including bladder, ovarian, and cervical cancers and head and neck squamous cell carcinoma (HNSCC) [6]. GP130 phosphorylation exposes the STAT3 binding site to induce STAT3 phosphorylation, after which it enters the nucleus and initiates transcription [11].

Bazedoxifene, approved by the FDA (Food and Drug Administration) as a selective estrogen modulator for osteoporosis treatment, is a new IL-6/GP130 inhibitor [16]. In addition to osteoporosis treatment, to date, it has been tested in breast cancer, blastoma, and pancreatic cancer [17,18,19,20]. However, no in vitro or in vivo studies on the effects of Bazedoxifene on cervical cancer have been reported to date. Therefore, we tested the potential inhibitory role of Bazedoxifene in cervical cancer.

In this study, the in vitro and in vivo anticancer effects of Bazedoxifene on human cervical cancer cells via the blocking of GP130 were investigated. Our results may provide a novel approach for using the GP130 inhibitor Bazedoxifene for the treatment of cervical cancer.

## 2. Results

### 2.1. Bazedoxifene Inhibited Proliferation of Cervical Cancer Cells

SiHa, CaSki, and HeLa cell lines were used to investigate the effect of Bazedoxifene on cervical cancer cell proliferation and cytotoxicity. To investigate cytotoxicity, cervical cancer cell lines were treated with varying concentrations of Bazedoxifene for 8 h. In SiHa, CaSki, and HeLa cells, the cell viability decreased significantly with a Bazedoxifene concentration of 4 μM (Figure 1A). The 50% inhibitory concentrations (IC_50_) for Bazedoxifene were calculated from the MTT cell survival curves for all cell lines. The IC_50_ values for the three cell lines were 3.79 μM (SiHa), 4.827 μM (HeLa), and 4.018 μM (CaSki) (Figure 1B). In addition, in order to confirm the inhibition of cell growth by Bazedoxifene, the growth of SiHa, HeLa, and CaSki cells was monitored for 0–72 h after treatment with Bazedoxifene concentrations of 1, 5, or 10 μM. In SiHa and HeLa cells, cell growth was inhibited at all concentrations of Bazedoxifene, and in CaSki cells, cell growth was inhibited at 5 and 10 μM Bazedoxifene (Figure 1C). In the HPV-negative cell line C33A, cytotoxicity and cell growth were measured with different Bazedoxifene concentrations and treatment durations. Contrary to the results of HPV-positive cell lines, there was no significant cytotoxicity or effect on cell growth (Appendix A).

### 2.2. Bazedoxifene Inhibits Cervix Cancer Cell Colony-Forming, Migration, and Invasion Ability

Colony formation, wound healing, and invasion assays were performed after Bazedoxifene treatment in cervical cancer cell lines. In SiHa, HeLa, and CaSki cells, cell migration was significantly reduced after 5 and 10 μM Bazedoxifene treatment (Figure 2A). The cell invasion assay showed that cell invasion decreased as the concentration of Bazedoxifene increased. In particular, the greatest inhibition of cell invasion and the smallest number of infiltrated cells were obtained with 10 μM Bazedoxifene (Figure 2B). In addition, Bazedoxifene concentrations of 5 and 10 μM were found to significantly decrease the colony-forming ability of cervical cancer cells. Bazedoxifene at a concentration of 10 μM resulted in the greatest decrease in the colony-forming ability (Figure 2C). As shown in Figure 2, treatment with Bazedoxifene reduced the migration, invasion, and colonization of cervical cancer cells in a concentration-specific manner.

### 2.3. Bazedoxifene Promotes Apoptosis in Cervix Cancer Cells

Apoptosis is an important process in tumor growth. Therefore, we assessed the effect of Bazedoxifene on cervical cancer cell death. In SiHa cells, a difference in apoptosis was confirmed after treatment with Bazedoxifene at 1, 5, and 10 μM for 48 h. The results confirmed that apoptosis of SiHa cells was increased at 48 h, with the magnitude of the increase depending on the concentration of Bazedoxifene. In particular, apoptosis increased the most after treatment with 10 μM Bazedoxifene (Figure 3A). Degree of apoptosis was converted into a percentage, and it was confirmed that the higher the concentration of Bazedoxifene, the higher the rate of apoptosis was 5, 8, and 12% (Figure 3B).

Next, Western blot analysis was performed to examine the changes in the apoptosis signal caused by Bazedoxifene. In SiHa cells, as the concentration of Bazedoxifene increased, the apoptosis markers Bim and Bax increased, and the expression of cell survival markers Mcl-1 and Bcl-xL decreased (Figure 4A,B).

### 2.4. Bazedoxifene Inhibits p-GP130/p-STAT3 and EMT Signaling

The ability of Bazedoxifene to inhibit GP130/STAT3 signaling was evaluated in SiHa, HeLa, and CaSki cervical cancer cells. Bazedoxifene dose-dependently reduced the expression of p-STAT3 and p-GP130 compared with hyper-IL-6 in SiHa, HeLa, and CaSki cells (Figure 4A). In addition, Bazedoxifene may affect other downstream targets of GP130. The results show that the expression of p-ERK1/2 was downregulated compared to hyper-IL-6 in SiHa, HeLa, and CaSki cells, and the magnitude of downregulation depended on the concentration of Bazedoxifene (Figure 5A). Additionally, we analyzed the effect on the EMT pathway, one of the sub-signals of Bazedoxifene. In SiHa, HeLa, and CaSki cells, the expression of E-cadherin was upregulated and the expression levels of β-catenin, vimentin, and Wnt5β were downregulated in a concentration-dependent manner (Figure 5B). This indicates that Bazedoxifene inhibits GP130/STAT3 via EMT signaling, a downstream signal in cervical cancer cells.

### 2.5. Bazedoxifene Suppresses Tumor Growth in a Dose-Dependent Manner and Inhibits EMT Progression

A xenograft mouse model was established using SiHa cervical cancer cells, and the tumor suppressor effect of Bazedoxifene was analyzed. When the size of the xenograft tumor was 500 mm^3^ or more, 1, 5, or 10 mpk Bazedoxifene was orally administered daily for 2 weeks to observe the changes. Four weeks later, the mice were sacrificed. The size and weight of tumors in mice treated with 1, 5, and 10 mpk Bazedoxifene were reduced compared to those in the control group (Figure 6A,B). The greatest differences in the sizes and weights of tumors were in the 10 mpk group. This indicates that 10 mpk Bazedoxifene can inhibit the growth of cervical cancer in vivo. The xenograft tumors obtained after the end of the experiment were not significantly different in 1 and 5 mpk groups compared to the control group, but there was a significant difference in the group that received 10 mpk (Figure 6C). The EMT-related signal, one of the sub-signals of Bazedoxifene, was confirmed in tissues obtained by xenotransplantation. The expression of E-cadherin was upregulated and the expression of β-catenin, vimentin, and Wnt5β was downregulated as the concentration of Bazedoxifene increased (Figure 6D). These data indicate that Bazedoxifene influences tumor growth by modulating EMT signaling.

## 3. Discussion

Cervical cancer, most commonly squamous cell carcinoma, is one of the most common malignancies in women [21]. Although prevention has increased due to the development of the cervical cancer vaccine, the incidence rate is still high in the Sahara Desert, South Africa, and South Asia [22]. In a variety of cancers, IL-6 is overexpressed and is associated with the activity of STAT3 [6]. In tumors, IL-6 serves multiple functions associated with inflammation and immune suppression by interacting with the surrounding matrix [23]. IL-6 in cervical cancer is a known potential biomarker that acts by inducing vascular epithelial growth factor-dependent angiogenesis in a STAT3-dependent manner and promoting tumor proliferation [24,25]. However, the contribution of IL-6 to STAT3 activity in cervical cancer is not well understood.

Blocking IL-6 and GP130 signaling in HPV-positive cervical cancer cells abolished STAT3 phosphorylation, and blocking the GP130 signal was found to be a major determinant of STAT3 phosphorylation [26]. As GP130 is located at the intersection of the oncogenic signaling network and is critical to network activation, blocking GP130 activity is an attractive treatment method for GP130-dependent cancer [27]. In several studies, anti-IL-6 blocking antibodies or sGP130Fc has been demonstrated to inhibit IL-6 signaling and induce apoptosis in breast and pancreatic cancer cells and animal models [28,29,30]. However, these antibody treatments have been observed to markedly elevate systemic IL-6 [31]. Therefore, it is highly desirable to identify alternative small-molecule drugs to target IL-6/GP130 signaling and provide new options for anticancer therapy. In order to overcome this problem, an inhibitor of GP130, an important part of the receptor signaling complex of IL-6/IL-6R/GP130, is required. The GP130 cytokine family regulates a wide range of processes that influence bone remodeling, cancer development, and metastasis through paracrine and autocrine mechanisms. Although the literature has defined numerous roles of members of the GP130 cytokine family to date, there are still unknown mechanisms of action [29]. Bazedoxifene is known to bind to GP130, inhibit downstream signaling proteins such as STAT3 and ERK, a sub-signal, and increase the activation of the apoptosis pathway. In this study, we found that treating the HPV-positive cervical cancer cells SiHa, HeLa, and CaSki with a GP130 inhibitor, Bazedoxifene, inhibited cell growth, cell invasion, and colony formation. According to the concentration of Bazedoxifene in SiHa, cervical cancer cells, Bim and Bax, increase markers of the apoptosis pathway, increased over time. Conversely, the expression of Mcl-1 and Bcl-xL, which are markers of reduction in the apoptosis pathway, decreased. Similar to previous studies that reported that Bazedoxifene inhibits cell viability, proliferation, and migration and promotes apoptosis of pancreatic cancer [20] and medulloblastoma cells [17], our in vitro study also confirmed that Bazedoxifene promotes apoptosis in cervical cancer.

In addition, the results show that STAT3 and ERK signals, which are sub-signals of p-GP130, were inhibited in SiHa, HeLa, and CaSki cervical cancer cells according to the concentration of Bazedoxifene. Invasion and metastasis of epithelial malignancies are associated with the acquisition of EMT characteristics, and tumor invasion is an important factor in reducing tumor cell adhesion and improving migration and motility [32,33]. Therefore, changes in EMT signaling proteins were assessed after treating SiHa, HeLa, and CaSki cervical cancer cells with Bazedoxifene at different concentrations. E-cadherin expression increased, and β-catenin, vimentin, and Wnt5β expression decreased according to the concentration of Bazedoxifene in SiHa, HeLa, and CaSki cells. In a previous study, treating ovarian cancer [27] and rhabdomyosarcoma cells [34] with anti-GP130 antibody or GP130 shRNA was found to inhibit STAT3 phosphorylation and/or cell survival, which is consistent with the results of this study. In addition, STAT proteins are known to be involved in cancer tumorigenesis and the promotion of cancer development, and among STAT proteins, STAT3 is known to induce EMT [35]. Similar to these studies, we also confirmed that STAT3 inhibition suppressed EMT.

In the in vivo experiment, xenograft mice were established by injecting SiHa cells into the flanks of nude mice, which are immunosuppressing mice, and treated with Bazedoxifene at different concentrations to suppress tumor growth. After the in vivo experiment, EMT signal transduction in mouse tissues was analyzed through Western blot analysis, and it was confirmed that the suppression of EMT signal transduction increased as the concentration of Bazedoxifene increased. Based on our results, we suggest that Bazedoxifene is a potent inhibitor of GP130/STAT3 signaling in HPV-positive cervical cancer cells. In cervical cancer, STAT3 has gained importance from the observation that its presence and activity are associated with the malignancy of cervical lesions [36]. High-risk HPV-positive cells display significantly larger amounts of active STAT3 compared to HPV-negative cells [37]. We have demonstrated through in vitro and in vivo experiments that Bazedoxifene blocks the activation of STAT3 in cervical cancer, which can inhibit cervical cancer growth and affect EMT signaling and apoptosis associated with cancer cell migration and invasion. Therefore, it is necessary to study the inhibition of STAT3 in HPV-positive cervical cancer cells, which was the focus of this study. In conclusion, the results suggest that Bazedoxifene is a potential treatment for human cervical cancer.

## 4. Materials and Methods

### 4.1. Cell Lines and Cell Culture

We purchased the cervical cancer cell lines SiHa (HPV16 positive), HeLa (HPV18 positive), and CaSki (HPV16 positive) from the Korea Cell Line Bank and the HPV-negative cell line C33A from the American Type Culture Collection (HTB-31, Rockville, MD, USA). For cell growth conditions, DMEM or RPMI 1640 medium was used, to which 10% *v*/*v* fetal bovine serum (FBS, Gicbo, Waltham, MA, USA) and 1% *v*/*v* penicillin–streptomycin (PS, Gibco, Waltham, MA, USA) mixture (source) were added. The cell lines were maintained at 37 °C in humidified air with 95% air and 5% CO_2_, and no cells exceeded 20 passages. HeLa cells were cultured in complete Dulbecco’s modified Eagle medium (DMEM, Welgene, Daegu, Korea). SiHa and CaSki cells were cultured in RPMI 1640 (Welgene, Daegu, Korea). C33A cells were cultured in MEM (Welgene, Daegu, Korea).

### 4.2. Cell Proliferation Assay

3-(4,5-Dimethylthiazolyl)-2,5-diphenyltetrazolium bromide (MTT) and cell counting kit-8 (CCK-8) assays were performed to assess cell proliferative capacity and cytotoxicity. After seeding 1 × 10^5^ cells/well in 48-well plates, cells were treated with 1, 5, or 10 μM Bazedoxifene (Sigma-Aldrich, Merck, Darmstadt, Germany) for 6 h. For the CCK-8 analysis, 50 μL of CCK-8 (CCK-8, Dojindo, Kumamoto, Japan) was added at 0, 24, 48, and 72 h and allowed to react for 1 h, and then the optical density (OD) value was measured. OD values of each well were measured using an ELISA microplate reader (Epoch, BioTek Instruments, Winooski, VT, USA) at a wavelength of 450 nm. For MTT analysis, 100 μL of MTT solution (5 mg/mL, Sigma-Aldrich, Merck, Darmstadt, Germany) was added to the wells for 4 h. The solution was then replaced with 500 μL of dimethyl sulfoxide (DMSO, Sigma-Aldrich, Merck, Darmstadt, Germany) and the OD value was measured at a 560 nm wavelength. The experiment was carried out with 3 or more repetitions. IC_50_ values were calculated from a log([drug]) versus normalized response curve fit using GraphPad Prism version 5.00 for Windows (GraphPad Software, San Diego, CA, USA).

### 4.3. Wound Healing Migration Assay

Cell migration was evaluated through wound healing analysis. The cells were seeded into a 6-well plate at a density of approximately 5 × 10^5^, treated with Bazedoxifene at concentrations of 1, 5, or 10 μM for 24 h, and allowed to grow to 100% confluence in complete medium. When the cell density reached 100%, a sterile 200 μL tip was used to scrape a single homogeneous layer to create a homogeneous wound. After making the homogeneous wound, the cells were washed with phosphate-buffered saline (PBS). Cell migration was monitored for 0, 24, and 48 h. The area of the scratch was measured using ImageJ software (NIH Image, Bethesda, MD, USA) and calculated as a percentage (width of zone 0 h/width of zone at 24 or 48 h). It was calculated by standardizing the area of the control cells. The migrated cells were counted in 9 fields under a 20× objective.

### 4.4. Matrigel Invasion Assay

Analysis of the Matrigel invasion chamber (pore size: 8 mm, BD Bioscience, Lowell, MA, USA) was performed according to the manufacturer’s protocol. After inoculating the cells in a 6-well plate at a density of 1 × 10^6^, they were treated with Bazedoxifene with 1, 5, or 10 μM for 24 h. After 24 h, 5 × 10^5^ cells were inoculated into the upper chamber with serum-free medium, and complete medium (10% FBS, 1% penicillin–streptomycin) was added to the lower chamber. Matrigel chambers were incubated for 48 h in an incubator set at 37 °C and 5% CO_2_. After 48 h, non-invasive cells on the top of the chamber were removed with a cotton swab. In order to determine the number of invading cells under the filter, the cells were stained using a dyeing reagent (Diff Quik, Kobe, Japan) and the stained cells were counted using ImageJ software.

### 4.5. Colony Formation

SiHa, HeLa, and CaSki cells were treated with 1, 5, or 10 μM Bazedoxifene for 48 h and then seeded for colony formation in 6-well plates at 1 × 10^5^ cells per well. After 2 weeks of incubation, the cells were fixed with cold 100% methanol for 5 min and stained with crystal violet (crystal violet containing 0.5% in phosphate-buffered saline (PBS), Samcheon, Gyeonggi-do, Korea) for 20 min. After staining, colony formation was observed under a microscope. The result of the colony formation analysis was quantified using ImageJ software.

### 4.6. Western Blot

SiHa, HeLa, and CaSki cells were inoculated into 6-well plates at a density of 5 × 10^5^ cells/well. After inoculation, cells were starved overnight, treated with 1, 5, or 10 μM Bazedoxifene for 8 h, and stimulated with 5 ng/mL hyper-IL-6 [38,39] for 5 min. Cells were harvested and reacted with a mixed solution of RIPA buffer and protease inhibitor cocktail 100× (Invitrogen, Carlsbad, CA, USA) for at least 40 min on ice and vortexed every 10 min. The lysate was centrifuged at 4 °C for 20 min at 13,000 rpm and the supernatant was transferred to a new tube. Equal amounts of protein (30 μg/lane) were separated with sodium dodecyl sulfate polyacrylamide gel electrophoresis (SDS-PAGE) and then transferred to polyvinylidene fluoride membranes (Clear Blot Membrane-P, Atto, Tokyo, Japan). After blocking with 5% skim milk or 5% bovine serum albumin (BSA), the membrane was incubated overnight at 4 °C with the targeted primary antibody. The next day, the membrane was washed 6 times in Tris buffer with 0.1% Tween 20 and incubated with secondary antibody for 1 h 30 min at room temperature. The Western blotting detection reagent used was the SuperSignal™ West Femto Maximum Sensitivity Substrate (Thermo Scientific, Waltham, MA, USA). β-actin level was used as loading control. The bands were visualized using the Chemi DOC XRS X system (Bio-Rad, Hercules, CA, USA), and the band intensities were quantified by ImageJ software.

### 4.7. Annexin V and Propidium Iodide (PI) Staining for Apoptosis Assay

Apoptosis was assessed using the flow cytometry of control cells stained with Annexin V-FITC and PI following the protocol of the Annexin V-FITC apoptosis detection kit (BD Bioscience, Franklin Lakes, NJ, USA). After inoculating 5 × 10^5^ SiHa cells in 6-well plates, the cells were treated with 1, 5, or 10 μM Bazedoxifene for 48 h at 37 °C and 5% CO_2_. The cells were then collected, washed with PBS (with 0.5–1% BSA), and resuspended in 500 μL of 1× Annexin-binding buffer. They were then incubated at room temperature with Annexin V-FITC and PI staining in the absence of light. After 10 min, the samples were immediately analyzed via flow cytometry.

### 4.8. Xenograft in Mouse

BALB/c nude mice (*n* = 4; 5 weeks old; Orient Bio, Seongnam, Korea) were maintained at constant temperature and humidity (Catholic University protocol) and all procedures were approved by the Institutional Animal Care and Use Committee of the Catholic University of Korea (approval number: CUK-IACUC-2019-026, permission code, 29 May 2019) and were compliant with the legal mandates and federal guidelines for the care and maintenance of laboratory animals. All experimental work was compliant with the legal mandates and federal guidelines for the care and maintenance of laboratory animals. Each mouse was injected subcutaneously with 100 μL of suspension of SiHa cells in the dorsal scapular area. When the tumor size was about 500 mm^3^, a week after cell injection, Bazedoxifene (1, 5, or 10 mg/kg) was administered once a day by gavage for a total of 10 days. Tumor size was measured once every 3 days using a caliper. Tumor volume was measured using a simplified equation to estimate spheroids (length × width × 0.5). Each tumor was harvested 14 days after Bazedoxifene treatment.

### 4.9. Statistics

All quantitative results are expressed as mean ± S.D. Statistically significant differences were identified using one-way analysis of variance (ANOVA) or a Student’s *t*-test. Only values with *p* < 0.05 were considered statistically significant.

## 5. Conclusions

Our research demonstrates that lncRNA SRA can act as an inhibitor of HPV-positive cervical cancer cell progression. In this study, we investigated the anticancer properties of Bazedoxifene in HPV-positive cervical cancer cells and confirmed that Bazedoxifene inhibits the GP130/STAT3 pathway and suppresses the EMT sub-signal. Our research can provide a basis for cervical cancer replacement therapy.

## Figures and Tables

**Figure 1 ijms-22-08693-f001:**
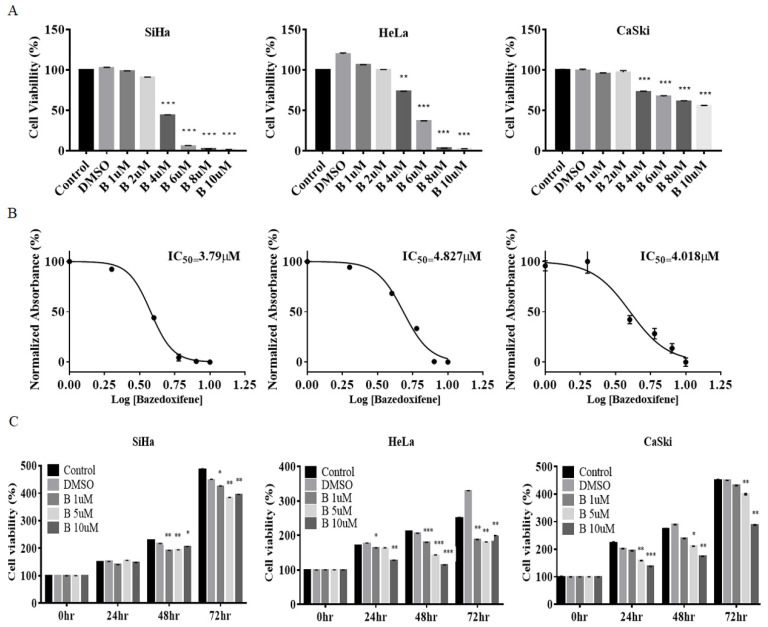
Bazedoxifene inhibited the viability of cervical cancer cells in a dose-dependent manner. (**A**) SiHa, HeLa, and CaSki cells were treated with Bazedoxifene at the indicated concentrations in triplicate for 48 h and the MTT assay was performed to analyze cell viability (*, *p* < 0.5, **, *p* < 0.1, ***, *p* < 0.001 vs. control). (**B**) IC_50_ values were measured for each cell line following 48 h of treatment with various concentrations (1, 2, 4, 6, 8, and 10 μM) of Bazedoxifene. (**C**) Cervical cancer cells were treated with Bazedoxifene (1, 5, or 10 μM) in triplicate for 0, 24, 48, or 72 h, and the CCK assay was performed to analyze cell viability (*, *p* < 0.05 **, *p* < 0.01, ***, *p* < 0.001 vs. control). B: Bazedoxifene.

**Figure 2 ijms-22-08693-f002:**
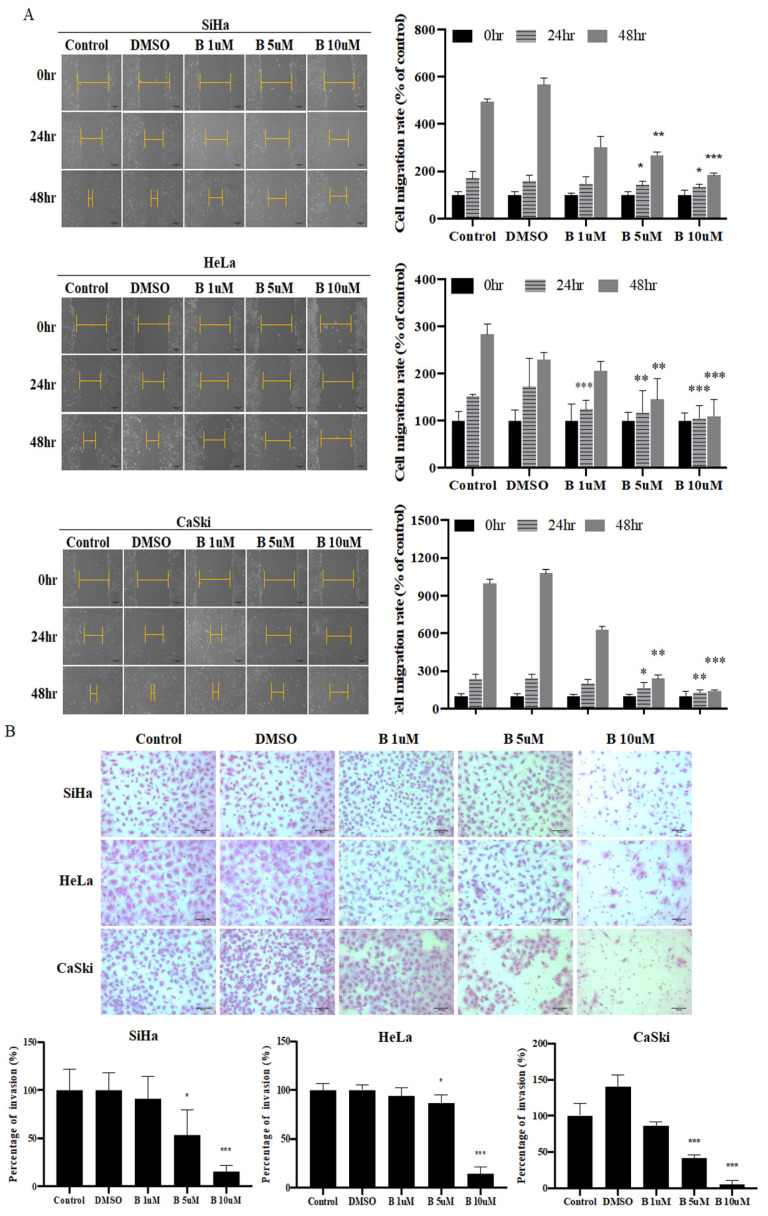
Bazedoxifene inhibited migration, invasion, and colony formation of cervical cancer cells. (**A**) Migration assay was used to determine migration of SiHa, HeLa, CaSki cells after 0, 24, and 48 h of Bazedoxifene treatment (*, *p* < 0.05; **, *p* < 0.01; ***, *p* < 0.001 vs. control). (**B**) Matrigel invasion assay was used to determine invasion of SiHa, HeLa, and CaSki cells after 48 h of Bazedoxifene treatment. (**C**) Colony formation assay was conducted on SiHa, HeLa, and CaSki cells treated with Bazedoxifene for 48 h. Clones were grown by reseeding at 1000 cells per well and incubating for 2–3 weeks. Scale bar, 100 μm. Bars indicate mean ± standard deviation of three independent experiments performed in triplicate (*, *p* < 0.05; **, *p* < 0.01; ***, *p* < 0.001 vs. control). B: Bazedoxifene.

**Figure 3 ijms-22-08693-f003:**
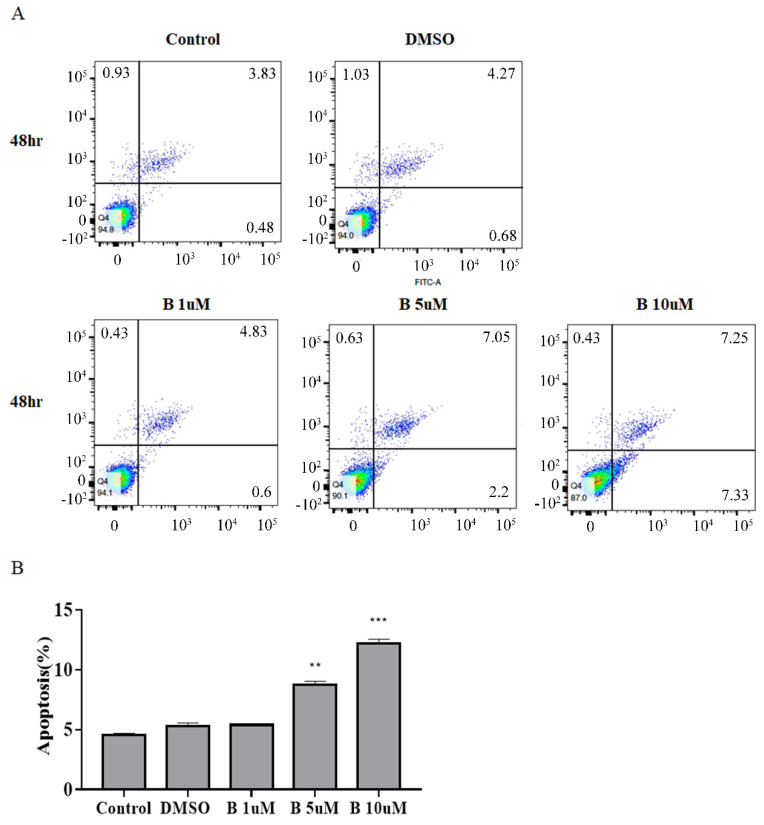
Bazedoxifene induces apoptosis in cervical cancer cells. (**A**) Apoptotic effects were detected using Annexin V-FITC/PI staining by flow cytometry in SiHa cervical cancer cells after 48 h. (**B**) Statistical analysis of the percentage of cell apoptosis. Bars indicate mean ± standard deviation of three independent experiments performed in triplicate (**, *p* < 0.01; ***, *p* < 0.001 vs. control). B: Bazedoxifene.

**Figure 4 ijms-22-08693-f004:**
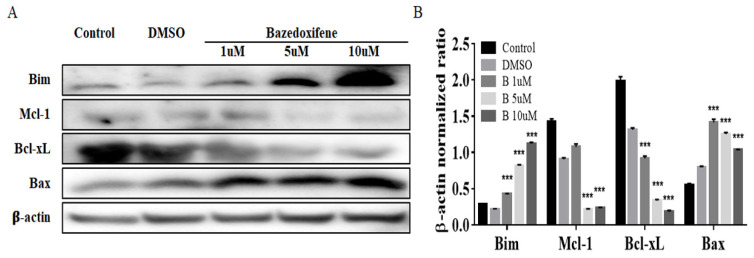
Expression analysis of apoptosis marker proteins in Bazedoxifene-treated SiHa cells. (**A**) Cells were treated for 48 h with the indicated doses of Bazedoxifene. Cell extracts prepared after treatment were resolved by SDS-PAGE and analyzed by Western blot analysis with antibodies against the indicated proteins. (**B**) The band intensities were quantitated. The loading control was β-actin. Bars indicate mean ± standard deviation of three independent experiments performed in triplicate (***, *p* < 0.001 vs. control). B: Bazedoxifene.

**Figure 5 ijms-22-08693-f005:**
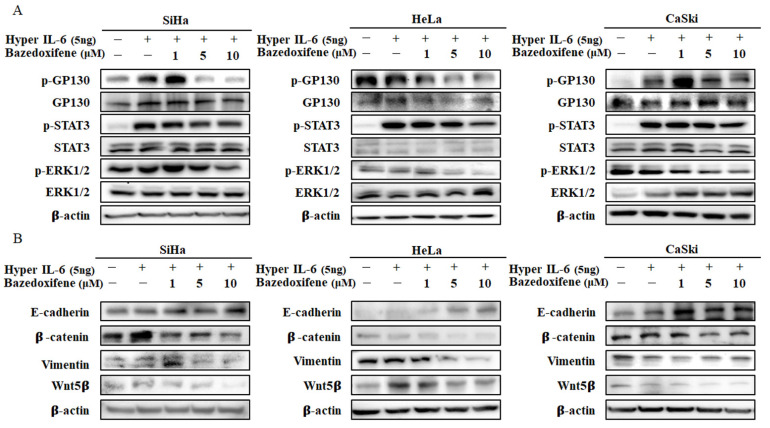
Bazedoxifene inhibited the expression of p-GP130, p-STAT3, p-ERK1/2, and EMT signaling in cervical cancer cells. (**A**) Bazedoxifene decreased the expression levels of p-GP130, p-STAT3, and p-ERK1/2 in SiHa, HeLa, and CaSki cells. (**B**) Bazedoxifene decreased the expression levels of EMT signaling proteins in SiHa, HeLa, and CaSki cells.

**Figure 6 ijms-22-08693-f006:**
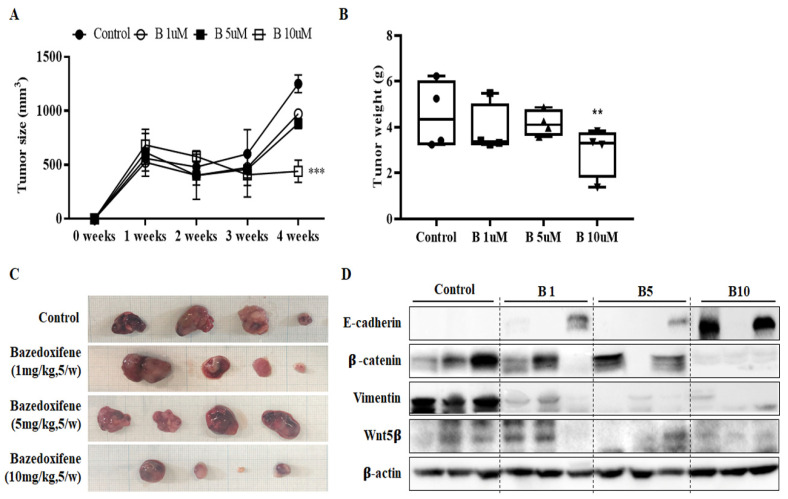
Bazedoxifene inhibited SiHa tumor growth in vivo. SiHa cells (1 × 10^7^) were injected subcutaneously into Balb/c nude mice with an equal volume of PBS. When tumors had grown to 150 mm^3^, mice were administered vehicle or 1, 5, or 10 mg/kg Bazedoxifene daily by oral gavage. (**A**) Tumor volumes were calculated from caliper measurements. (**B**) After 14 days of treatment, all mice were sacrificed, and the total mass of each tumor was determined at autopsy (*n* = 4 mice per treatment group). (**C**) The tumor masses were excised for comparison between groups. (**D**) The phosphorylation levels of EMT signaling proteins were determined using Western blot analysis of the harvested tumor tissue. β-actin served as a loading control (**, *p* < 0.01 ***, *p* < 0.001 vs. control). B: Bazedoxifene.

## Data Availability

Not applicable.

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
