# Peer review of "Bazedoxifene, a GP130 Inhibitor, Modulates EMT Signaling and Exhibits Antitumor Effects in HPV-Positive Cervical Cancer"

_ijms, 2021, doi:10.3390/ijms22168693_

Round 1
Reviewer 1 Report
The manuscript presented some interesting findings. The introduction provides background and includes relevant references. The research design is appropriate. Only minor editing of English language are required.
In my opinion minor revision is required.
Kind regards
Some points were listed below
Introduction
Lines 46-54: please, give information whether STAT3 is active in phosphorylated or de phosphorylated form – it is essential to understand the molecular action of the drug
Lines 56-57: “Bazedoxifene, approved by the FDA as a selective estrogen modulator for osteoporosis treatment, is a new IL-6/GP130 inhibitor.” – please, add appropriate reference here
Results
Fig 1 B and line 257: were the following concentrations 1, 5, or 10μM of Bazedoxifene used for IC 50 calculations? Or there were more points? Line 87: “various concentrations” – please, be precise which ones
Fig 2 Y axis “number of colonies”?
Fig 3A – please, make it clear, which result is after 24 h and which 48 h of treatment.
Discussion
It may be interesting to compare the data obtained using HPV containing cell lines with HPV negative cell line C33A. The Authors decided not to include the results since Bazedoxifene had only minor effects on C33A proliferation
Lines 206-8: “According to the concentration of Bazedoxifene in SiHa, cervical cancer cells, Bim and Bax, markers of the apoptosis pathway, increased over time, and Mcl-1, and Bcl-xL decreased” the Authors may split the sentence in two to make it more clear
Materials and Methods
Please specify the particular numbers/collections of cell lines used (possibly according to ATCC collection).
You may change line 247 for: “10% v/v fetal bovine serum (FBS, Gicbo, USA) and 1% v/v penicillin streptomycin mixture (source) were added to cell culture medium.”
Line 249: please, provide name of the manufacturer for “penicillin streptomycin”
Line 256: please, change “confirm” for “assess”\
4.2. Cell proliferation assay:
Were any reference wavelengths used for assays (especially for MTT?)
Please, add information how IC 50 value was calculated
Line 299: Why the cells were starved overnight for protein expression experiments?
Lines 304-5: “A total of 30μg of protein was transferred to a polyvinylidene difluoride membrane through 10% SDS-PAGE gel separation” – the meaning is elusive, please be more specific
Line 309: please, specify what kind of results (densitometry?)
Animals
If possible, please give specific details of approval by Ethics Committee (permission number etc.) here in Materials and Methods section
Conclusions
Line 335: please change “Our research demonstrates that lncRNA SRA can act as an inhibitor of HPV-positive cervical cancer progression” for “Our research demonstrates that lncRNA SRA can act as an inhibitor of HPV-positive cervical cancer cell progression”
text editing:
Line 11: font height “ primary”
Line 58, 62, 63: please, use italics for “in vitro” „in vivo” “via” throughout the text
Line 281; “…they were Bazedoxifene treated with 1, 5, or 10 μM Bazedoxifene”?
Line 22: please change “but also provide a basis for cervical cancer replacement therapy.” for “but also may provide a basis for cervical cancer replacement therapy.”
Ref. numbers, e.g. line 28
Author Response
International Journal of Molecular Sciences
Responses to the reviewer’s comments:
We appreciate the opportunity to revise our manuscript. We had carefully considered all comments of the reviewers, and tried to describe in detail and correct all requiring corrections. We hope that this revision is sufficiently improved the paper will be accepted for publication in International Journal of Molecular Sciences.
We offer responses to the reviewer comments:
Reviewer #1 (Comments to the Authors (Required)):
- Introduction
Lines 46-54: please, give information whether STAT3 is active in phosphorylated or de phosphorylated form – it is essential to understand the molecular action of the drug
∴ Per the reviewer's suggestion, the activity of STAT3 was explained, and it was added in revised manuscript line 55 that STAT3 (P-STAT3) exerts a tumor-promoting effect through phosphorylation.
Lines 56-57: “Bazedoxifene, approved by the FDA as a selective estrogen modulator for osteoporosis treatment, is a new IL-6/GP130 inhibitor.” – please, add appropriate reference here
∴ Per the reviewer's suggestion, we have references to FDA approval of bazedoxifene were added to the revised manuscript in reference number 16.
- Results
Fig 1 B and line 257: were the following concentrations 1, 5, or 10μM of Bazedoxifene used for IC50 calculations? Or there were more points? Line 87: “various concentrations” – please, be precise which ones
∴ We thank the reviewer for these constructive suggestions, bazedoxifene used for IC50 calculations concentration point were 1, 2, 4, 6, 8 and 10uM. The revised content is indicated line 93 of the manuscripts.
Fig 2 Y axis “number of colonies”?
∴ We have corrected Figure 2 Y axis to ‘Relative colony area (%)’.
Fig 3A – please, make it clear, which result is after 24 h and which 48 h of treatment.
∴ We apologize for typographical errors. The results in Figure 3A are after 48 hours of treatment. We have corrected line 133 and indicated in the revised manuscript.
- Discussion
It may be interesting to compare the data obtained using HPV containing cell lines with HPV negative cell line C33A. The Authors decided not to include the results since Bazedoxifene had only minor effects on C33A proliferation
Lines 206-8: “According to the concentration of Bazedoxifene in SiHa, cervical cancer cells, Bim and Bax, markers of the apoptosis pathway, increased over time, and Mcl-1, and Bcl-xL decreased” the Authors may split the sentence in two to make it more clear
∴ Per the reviewer's suggestion, we divided the sentence into two and corrected on line 222 were indicated in the revised manuscripts.
- Materials and Methods
Please specify the particular numbers/collections of cell lines used (possibly according to ATCC collection).
∴ We changed ATCC collection of cell line information and corrected on line 262 were indicated in the manuscripts.
You may change line 247 for: “10% v/v fetal bovine serum (FBS, Gicbo, USA) and 1% v/v penicillin streptomycin mixture (source) were added to cell culture medium.”
∴ Per the reviewer's suggestion, we have corrected on line 263 and indicated in the revised manuscript.
Line 249: please, provide name of the manufacturer for “penicillin streptomycin”
∴ We added manufacturer for penicillin streptomycin and corrected on line 264 were indicated in the manuscripts.
Line 256: please, change “confirm” for “assess”
∴ We changed line 256 of the manuscript was revised and the corrections on line 272 were indicated in the manuscripts.
4.2. Cell proliferation assay:
Were any reference wavelengths used for assays (especially for MTT?)
∴ We thank the reviewer for these constructive suggestions, the reference wavelength for MTT analysis was 650nm and the reference wavelength for CCK analysis was 600nm.
Please, add information how IC50 value was calculated
∴We added IC50 calculated method and the corrections on line 282 were indicated in the manuscripts.
Line 299: Why the cells were starved overnight for protein expression experiments?
∴ We thank the reviewer for these constructive suggestions, all cells were equalized to the same phase of the cell cycle, and since serum affects various things such as cell growth and signaling pathway, cells were starved for 24 hours to minimize this.
Lines 304-5: “A total of 30μg of protein was transferred to a polyvinylidene difluoride membrane through 10% SDS-PAGE gel separation” – the meaning is elusive, please be more specific
∴ We revised western blot method and the corrections on line 321 were indicated in the manuscripts.
Line 309: please, specify what kind of results (densitometry?)
∴ We corrected the sentence and indicated the correction in line 327 in the revised manuscript.
Animals
If possible, please give specific details of approval by Ethics Committee (permission number etc.) here in Materials and Methods section
∴ We thank the reviewer for these constructive suggestions. We attach a separate file of information related to animal ethics (Catholic university IACUC report of institutional review board). All procedures were approved by the Institutional Animal Care and Use Committee of Catholic University of Korea (approval number: CUK-IACUC-2019-026) and were compliant with the legal mandates and federal guidelines for the care and maintenance of laboratory animals.
Conclusions
Line 335: please change “Our research demonstrates that lncRNA SRA can act as an inhibitor of HPV-positive cervical cancer progression”for“Our research demonstrates that lncRNA SRA can act as an inhibitor of HPV-positive cervical cancer cell progression”
∴ Per the reviewer's suggestion, we revised conclusion and the corrections on line 359 were indicated in the manuscripts.
text editing:
Line 11: font height “primary”
∴ We revised font and the corrections on line 11 were indicated in the manuscripts.
Line 58, 62, 63: please, use italics for “in vitro” „in vivo” “via” throughout the text
∴ Per the reviewer's suggestion, all mentioned words have been corrected in italics, and the corrections are indicated in the revised manuscript.
Line 281; “…they were Bazedoxifene treated with 1, 5, or 10 μM Bazedoxifene”?
∴We corrected the sentence (line 298).
Line 22: please change “but also provide a basis for cervical cancer replacement therapy.” for “but also may provide a basis for cervical cancer replacement therapy.”
∴ Per the reviewer's suggestion, we revised and the corrections on line 22 were indicated in the manuscripts.
Ref. numbers, e.g. line 28
∴ We revised reference and the corrections were indicated in the manuscripts.

Reviewer 2 Report
In the introduction, I would add an explanation regarding inflammation and other pro-inflammatory cytokines besides IL-6. Because this paragraph, which goes from cervical cancer incidence to IL-6, is illegibleIn my opinion, the results are very good. I would only separate the caption a and b under the third figure, because too much information is given there, which makes it difficult for the reader to know what is currently being described. . In my opinion, the discussion should be reorganized and expanded. There is no consideration and discussion of in-vitro and in vivo results, no mixing of these studies.
Author Response
Reviewer #2 (Comments to the Authors (Required)):
In the introduction, I would add an explanation regarding inflammation and other pro-inflammatory cytokines besides IL-6. Because this paragraph, which goes from cervical cancer incidence to IL-6, is illegible.
∴ Per the reviewer's suggestion, we revised introduction and the corrections on line 35 were indicated in the manuscripts. The reference is the same as the reference in the later content, so we did not add it separately.
In my opinion, the results are very good. I would only separate the caption a and b under the third figure, because too much information is given there, which makes it difficult for the reader to know what is currently being described.
∴ We thank the reviewer for these constructive suggestions. For the third figure, I modified it to figure 3 and figure 4 by splitting the captions. We marked the correction in the revised manuscripts.
In my opinion, the discussion should be reorganized and expanded. There is no consideration and discussion of in-vitro and in vivo results, no mixing of these studies.
∴ Per the reviewer's suggestion, we revised discussion and the corrections were indicated in the manuscripts.